# Optical Temperature Sensors Based on Down-Conversion Nd³⁺,Yb³⁺:LiYF₄ Microparticles

Anna Ginkel [1], Maksim Pudovkin [1,*], Ekaterina Oleynikova [1], Slella Korableva [1] and Oleg Morozov [1,2]

1 Institute of Physics, Kazan Federal University, 18th Kremlyovskaya Street, 420008 Kazan, Russia
2 Zavoisky Physical-Technical Institute, FRC Kazan Scientific Center of RAS, Sibirskii Ave., 10/7, 420029 Kazan, Russia
* Correspondence: jaz7778@list.ru

**Abstract:** $Nd^{3+}$ (0.3 mol.%), $Yb^{3+}$ (0, 1, 2, 3 and 5 mol.%): $LiYF_4$ phosphors were grown by the Bridgman–Stockbarger technique. The luminescence intensity ratio (LIR) of $Nd^{3+}$ ($^4F_{3/2}$–$^4I_{9/2}$, ~866 nm) and $Yb^{3+}$ emission ($^2F_{5/2}$–$^2F_{7/2}$, ~980 nm) was taken as a parameter. The energy exchange between $^4F_{3/2}$ ($Nd^{3+}$) and $^2F_{5/2}$ ($Yb^{3+}$) occurs via phonons, which elucidates the LIR temperature dependence. The influence of the cross-relaxation process on the temperature sensitivity was estimated as negligible. The LIR function depends on the $Yb^{3+}$ concentration at a fixed 0.3 mol.% $Nd^{3+}$. The maximum $S_a$ and $S_r$ value were reached for $Nd^{3+}$ (0.3%), $Yb^{3+}$ (1.0%): $LiYF_4$ ($S_a$ = 0.007 $K^{-1}$ at 320 K) and $Nd^{3+}$ (0.3%), $Yb^{3+}$ (5.0%): $LiYF_4$ ($S_r$ = 1, 1.03%*$K^{-1}$ at 260 K), respectively.

**Keywords:** luminescent thermometry; down-conversion; optical temperature sensors

## 1. Introduction

Currently, there is a rapid development of such branches of science and industry as circuitry, space and aviation industries, cell biology and theranostics. For these industries, there is an urgent need to measure the temperature fields of an object with submicron resolution. For some medical and scientific applications, non-contact or semi-contact temperature control methods with high spatial resolution are required [1,2]. An alternative method of temperature measurement is luminescent thermometry using phosphors having temperature-dependent luminescence parameters. Indeed, the ability to work in the UV, visible and near-IR spectral ranges, as well as nano-sized dimensionality of phosphors, provide submicron spatial resolution in temperature mapping. Here the phosphors serve as probes transmitting information about temperature over a distance through luminescence signal. In the case of temperature mapping of micro-circuit, the dielectric layer should be coated on its surface. In the case of hyperemia of the cell temperature mapping, the optical probes should be located in the special part of the studied object. For this application, some double-doped rare-earth inorganic phosphors demonstrate temperature-dependent spectral-kinetic properties due to temperature-dependent energy exchange processes between the doping ions [3,4]. A promising down-conversion ion pair is $Nd^{3+}$/$Yb^{3+}$. Particularly, the transfer of energy from $^4F_{3/2}$ ($Nd^{3+}$) level to $^2F_{5/2}$ ($Yb^{3+}$) one is accompanied by the emission of phonon [5]. The efficiency of such energy transfer depends on the temperature. This fact paves the way for the optical temperature measurement based on rare-earth double-doped phosphors. However, there is another interesting energy transfer process called cross-relaxation [6,7]. For example, cross-relaxation in the $Nd^{3+}$,$Yb^{3+}$:$LiYF_4$ system is possible under $Nd^{3+}$ excitation at 355 nm (($^2F_{7/2}$–$^2F_{5/2}$ ($Yb^{3+}$) and $^2K_{15/2}$/$^4G_{11/2}$–$^4F_{3/2}$ ($Nd^{3+}$)). However, the contribution of the cross-relaxation in the temperature dependence of spectral-kinetic characteristics is still questionable. Indeed, the work [8] informs that resonant cross-relaxation processes in $Ho^{3+}$:$LiYF_4$ are independent of the temperature. Since the process of cross relaxation can proceed with the participation

of phonons, which makes this mechanism temperature dependent, with a tendency to decrease the influence of cross relaxation with decreasing temperature. However, there is a cross-relaxation process without the participation of phonons, with resonant energy transfer depending on the distance between the suitable energy levels [9,10]. To investigate this contribution, we studied spectral characteristics of $Nd^{3+},Yb^{3+}:LiYF_4$ microparticles at two different regimes of optical excitation (at 355 and 520 nm). The choice of $LiYF_4$ matrix is based on its low phonon energy (~140 to 570 $cm^{-1}$) [11,12] that provides higher quantum luminescence yield due to a decrease in the probability of multiphonon nonradiative relaxation [13]. Additionally, $LiYF_4$ ensures the $Y^{3+}$ substitution without changing the valence [14]. For optical temperature sensing, essential characteristics of sensors are absolute and relative temperature sensitivities ($S_a$ and $S_r$, respectively), which are determined in Ref. [2]. Indeed, $Nd^{3+},Yb^{3+}:YF_3$ ($S_a$ = 0.002 $K^{-1}$ at 150 K) [15], $Nd^{3+},Yb^{3+}: YVO_4$ ($S_r$ = 0.085% $K^{-1}$ at 183 K) [16] demonstrate a noticeable temperature sensitivity. Here, the luminescence intensity ratio (LIR) between the $Nd^{3+}$ and $Yb^{3+}$ emissions was chosen as temperature-dependent parameter. The choice of LIR is justified by the fact that LIR is independent of the fluctuations in the of the excitation irradiation power density on opposite to luminescence intensity parameter. Thus, LIR has a great advantage in terms of "luminescence intensity", which depends on the power density of the exciting radiation. In the case of comparison of up- and down-conversion, it should be noted, that up-conversion phosphors are studied deeper compared to down-conversion ones. Moreover, in such ion pair as Er/Yb, Pr/Yb, Tm/Yb temperature sensitivity is achieved because of the presence of thermally coupled electron levels. It seems that it is difficult to manipulate the electron level structure and as a consequence temperature sensitivity. On the other hand, temperature sensitivity of down-conversion phosphors is mostly based on phonon-assisted energy exchange processes between doping ions. There are more ways to optimize the impact of these processes on the temperature sensitivity via choosing concentration of both ions and choosing matrices [17–19], In the case of matrix choice, fluoride matrices seem to be very promising compared to oxide ones due to low phonon energy that leads to the decrease in multiphonon relaxation probability. On the other hand, the $Nd^{3+}/Yb^{3+}$ ion pair is very specific due to relatively large difference in ionic radii. Since the $Nd^{3+}$ concentration was demonstrated to be low (0.1–0.5%) and $Yb^{3+}$ concentration should be notably higher (up to 10%) [18], the main requirements are imposed on the matrix ion that is expected to be substituted by $Yb^{3+}$ ion in order to create suitable concentration without miro-strains and double-phase formation. Here $Y^{3+}$ based matrices seem to be very promising. Such matrix, as $YF_3$ demonstrates undesirable broad emission related to the presence of specific defects under UV excitation [15,20]. In its turn, $LiYF_4$ is free from such peculiarities [21].

The objective of this work was to reveal the impact of excitation conditions on temperature sensitivity of $Nd^{3+},Yb^{3+}:LiYF_4$ phosphors.

## 2. Materials and Methods

Powders of LiF (99.999% purity, Lanhit, Russia), $YF_3$ (99.999% purity, Lanhit, Russia), NdF3 (99.999% purity, Lanhit, Russia) and $YbF_3$ (99.999% purity, Lanhit, Russia) fluorides were used as initial materials. The $YF_3$, $NdF_3$ and $YbF_3$ components were previously dried for 5 h at 100°C in vacuum. Then they were purified and fluorinated with 2 wt% $PbF_2$ for 5 h in vacuum at 900 °C and then were held for 2h in vacuum at a temperature above the corresponding melting points. Crystal growth process was carried out in a vacuum (~10–5–10–6 mbar) in a graphite crucible on a seed. The crystals were grown from the melt by vertical technique of Bridgman–Stockbarger. The temperature gradient at the solid-liquid interface was in the 85–100 °C/cm range. The pulling rate was chosen as 1 mm/h. The crystal size was 16 mm in length and 6 mm in diameter. A plate 2 mm thick was cut from the lower part of the grown crystal and was milled in the agate mortar into powder [19]. The phase composition was studied by X-ray diffraction (XRD) using a Shimadzu XRD-7000S X-ray diffractometer (Cu $K_\alpha$ radiation λ = 0.15406 nm). The luminescence spectra were detected via a CCD spectrometer "StellarNet" having spectral resolution of 0.5 nm. The

luminescence measurements were acquired using JV LOTIS TII laser system (420–1200 nm range, model LS-2134UTF) at 520 nm in the 80–320 K temperature range ($\lambda_{ex}$ = 520 nm and 355 nm, corresponds to $^4I_{9/2}$–$^2K_{13/2}$/$^2G_{9/2}$ and $^4I_{9/2}$–$^4D_{5/2}$, absorption bands of $Nd^{3+}$ ions, respectively). The measurements were carried out in the temperature range from 80 to 320 K. The temperature controlling was carried out via "CRYO industries" cooler equipped with LakeShore Model 325 temperature controller. The liquid nitrogen was utilized as a cooler. The pulse width and the rate of the pulse repetition were 10 ns and 10 Hz, respectively. Pulse laser irradiation was taken in order to exclude heating of the sample. The principal experimental set-up is represented in Figure S1 (Supplementary).

### 3. Results and Discussion

The sample phase composition was confirmed via method of X-ray diffraction (XRD). In particular, the XRD pattern of the $Nd^{3+}$ (0.3 mol.%),$Yb^{3+}$(1.0 mol.%):$LiYF_4$ sample is shown in Figure 1.

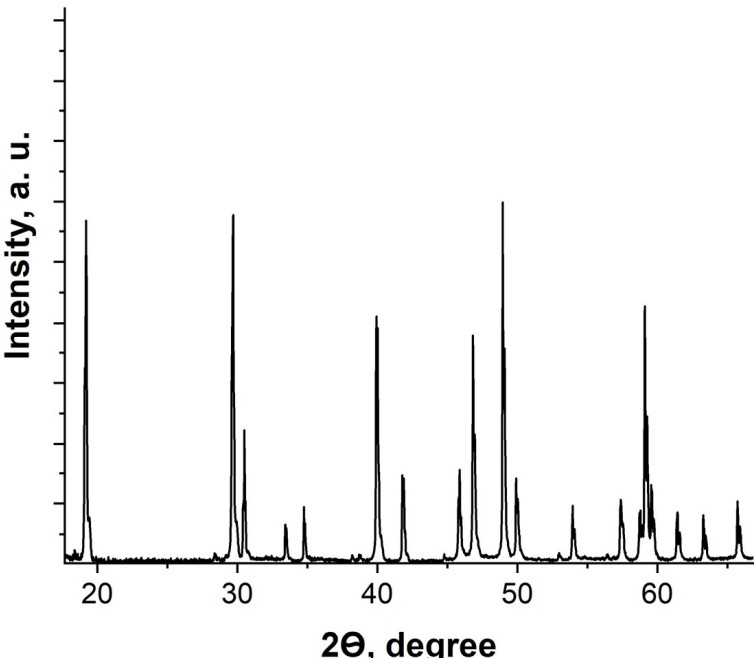

**Figure 1.** XRD patterns of $Nd^{3+}$ (0.3%), $Yb^{3+}$ (2.0%): $LiYF_4$ phosphors.

The X-ray diffraction pattern is consistent with the literature data [22] and corresponds to the trigonal structure of $LiYF_4$. The well-defined $LiYF_4$ peaks, the absence of impurity and amorphous phases are clearly seen. The Figure 2 demonstrates a simplified energy level diagram as well as main energy transfer processes [5]. These energy transfer processes involve $^4F_{3/2}$ ($Nd^{3+}$) and/or $^2F_{5/2}$ ($Yb^{3+}$) levels.

Particularly, under 355 nm excitation ($^4I_{9/2}$–$^4D_{5/2}$ absorption band of $Nd^{3+}$) the $^4F_{3/2}$ level of $Nd^{3+}$ in populated via cross-relaxation ($^2F_{7/2}$–$^2F_{5/2}$ ($Yb^{3+}$) and $^2K_{15/2}$–$^4F_{3/2}$ ($Nd^{3+}$)), radiative and nonradiative transitions from the higher energy levels. In turn, excitation at 520 nm ($^4I_{9/2}$–$^2K_{13/2}$/$^2G_{9/2}$ absorption band of $Nd^{3+}$) excluded cross-relaxation process. Indeed, in order to confirm the presence of radiative and nonradiative transitions, we detected the luminescence spectrum of $Nd^{3+}$ (0.3 mol.%) $Yb^{3+}$ (2 mol.%):$LiYF_4$ phosphors in the slightly broader (spectral range 710–1100 nm) spectral range under 355 nm excitation (Figure 3). According to the literature data [5], there were two peaks at ~740 and 800 nm that were interpreted as transitions from the upper $^4D_{5/2}$ and $^4P_{3/2}$ levels to $^4F_J$ of $Nd^{3+}$. Hence, the $^4F_J$ levels were populated via these radiative transitions. The nonradiative transitions between the $^4F_J$ levels were also possible and the population of the lowest $^4F_{3/2}$ level could also be nonradiative (for example from $^4F_{7/2}$).

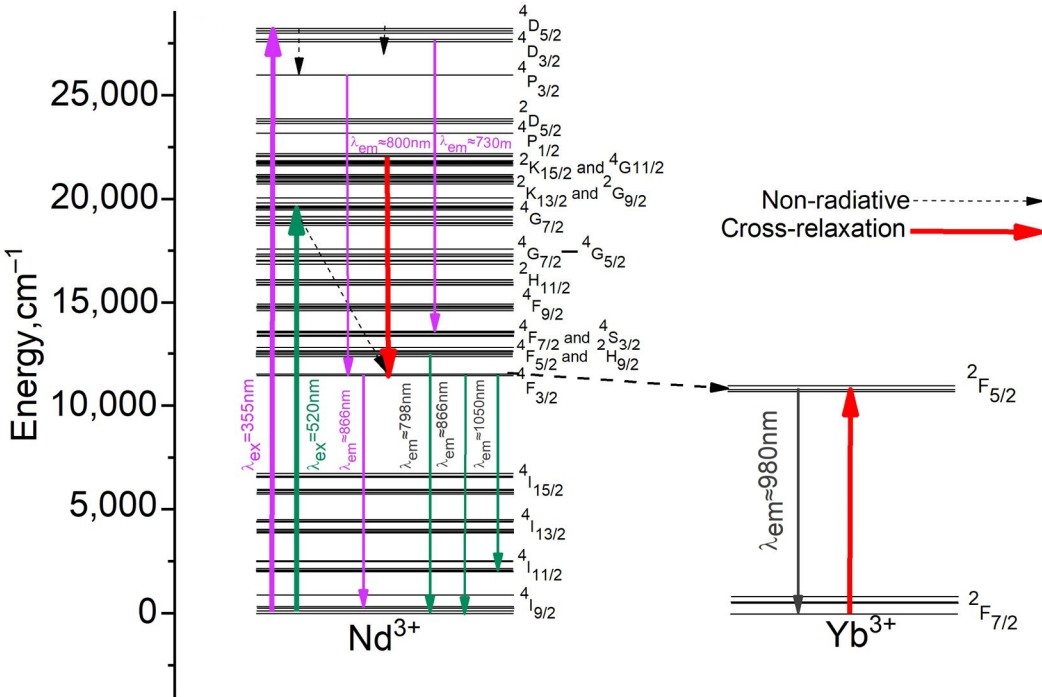

**Figure 2.** An energy level diagram of $Nd^{3+}$, $Yb^{3+}$ system.

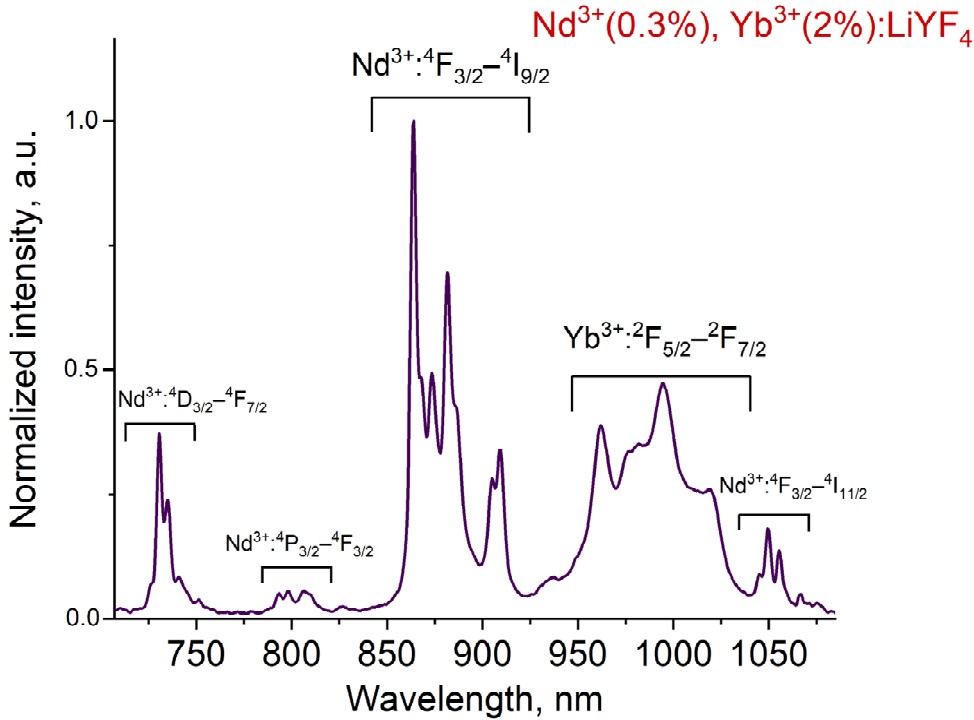

**Figure 3.** The room temperature spectrum of $Nd^{3+}$ (0.3%), $Yb^{3+}$ (2.0%): $LiYF_4$ phosphors normalized at 866 nm. The excitation $\lambda_{ex}$ = 355 nm corresponds to $^4I_{9/2}-^4D_{5/2}$ of $Nd^{3+}$ absorption band. The spectrum illustrates the presence of $^4D_{5/2}$, $^4P_{3/2}-^4F_J$ transitions.

Normalized luminescence spectra of $Nd^{3+}$ (0.3%), $Yb^{3+}$ (1.0–5.0 mol.%):$LiYF_4$ phosphors detected at room temperature are shown in Figure 4. Excitation of the system was carried out at $\lambda_{ex}$ = 520 nm, ($^2K_{13/2}-^2G_{9/2}$, absorption band of $Nd^{3+}$ ions). The observed $Yb^{3+}$ luminescence indicates the energy transfer from $Nd^{3+}$ to $Yb^{3+}$. All peaks were identi-

fied as the transitions from $^4F_{3/2}$ ($Nd^{3+}$) and $^2F_{5/2}$ ($Yb^{3+}$) to the lower energy levels. The peaks had a complex structure because of the complicated Stark structure of the electron levels. The emission intensity of $Yb^{3+}$ rises with the increase in $Yb^{3+}$ concentration compared to $Nd^{3+}$ emission intensity. These processes of the energy transfer involve $^4F_{3/2}$ ($Nd^{3+}$) and $^2F_{5/2}$ ($Yb^{3+}$).

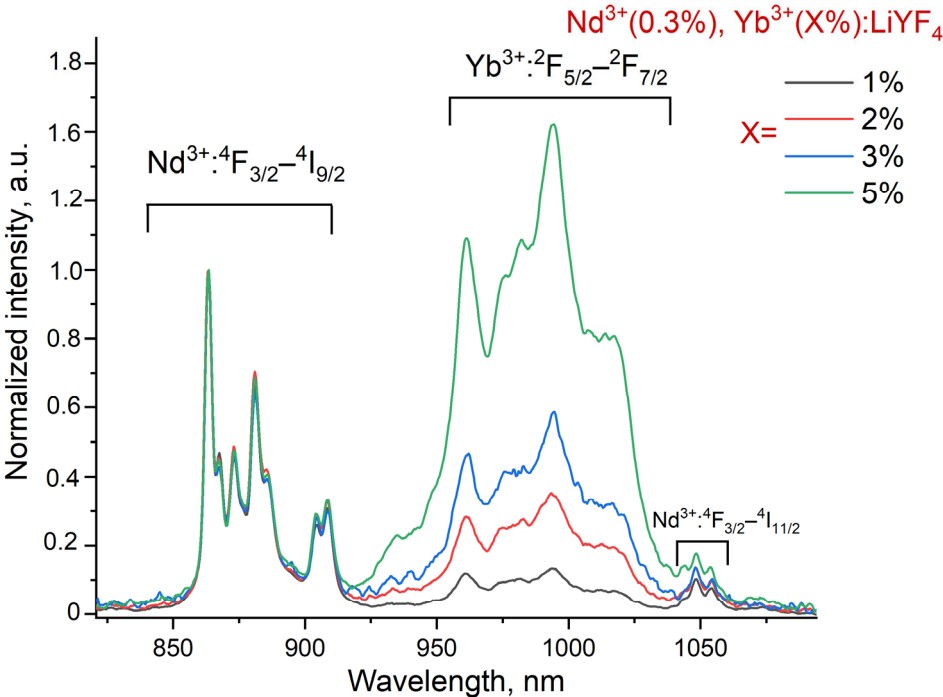

**Figure 4.** Normalized luminescence spectra of $Nd^{3+}$ (0.3%), $Yb^{3+}$ (1.0, 2.0, 3.0, and 5.0%):LiYF$_4$ phosphors recorded at room temperature. The excitation $\lambda_{ex} = 520$ nm corresponds to $^4I_{9/2}-^2K_{13/2}/^2G_{9/2}$ of $Nd^{3+}$ absorption band. The spectra are normalized at ~861 nm peak of $Nd^{3+}$.

According to ref. [18], the energy transfer between doping ions can be described by a process characterized by its probabilities. Energy exchange process between $Nd^{3+}$ ions seemed to be also probable; however, the $Nd^{3+}$ (0.3 mol.%) concentration for the studied samples did not change, its effect on the studied temperature-dependent luminescent properties was assumed to be negligible. Additionally, in ref [19,23], a cross-relaxation process was proposed for the $Nd^{3+}/Yb^{3+}$ pair. This process involves the $Nd^{3+}$ ($^2K_{15/2}/^4G_{11/2}-^4F_{3/2}$) and $Yb^{3+}$ ($^2F_{7/2}-^2F_{5/2}$) transitions. The room temperature luminescence decay curves of the $^4F_{3/2}-^4I_{9/2}$ transition (866 nm) of $Nd^{3+}$ under 355 nm excitation are shown in Figure 5a.

As we can see, with an increase in the $Yb^{3+}$ concentration, the $Nd^{3+}$ luminescence lifetime decreased, which indicates an energy transfer from $Nd^{3+}$ to $Yb^{3+}$. In addition, as mentioned above, the additional population of $Nd^{3+}$ and $Yb^{3+}$ levels may be related to cross-relaxation. Indeed, the luminescence rising time curve at an excitation wavelength of 355 nm are seen in Figure 5a. The luminescence rising time was shortened with an increase in $Yb^{3+}$ concentration; this process of shortening of the luminescence rising time curves indicates cross-relaxation. Indeed, the cross-relaxation was notably faster compared to radiative transitions from higher levels to $^4F_{3/2}$ one. Indeed, the work [24] estimated time resonant cross-relaxation between $Tm^{3+}$ ions from 8 μs (at 1.0% concentration) to 30 μs (at 4.0 % concentration). The contribution of faster cross-relaxation process in population of $^4F_{3/2}$ level increased with the increase in $Yb^{3+}$ that led to rising-time shortening. Figure 5b shows the luminescence decay time at an excitation wavelength of 520 nm as a function of $Yb^{3+}$ concentration. We can observe the same tendency for the decay curves. However, the rising time curves are not observed, hence, we excluded the cross-relaxation process by

using 520 nm excitation regime. It can also be seen that cross-relaxation was impossible for 520 nm photons, as they had too low energy (Figure 2).

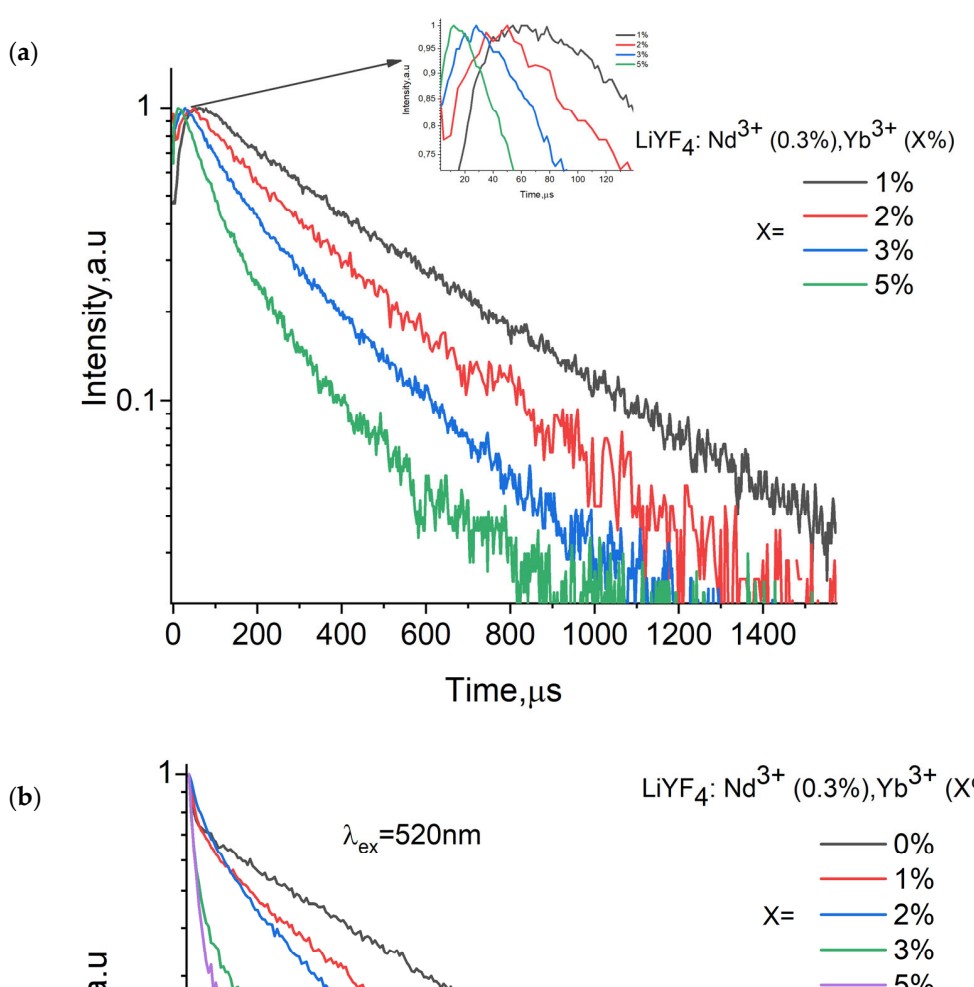

**Figure 5.** (**a**). Normalized room temperature lifetime curves of $^4F_{3/2}$–$^4I_{9/2}$ emission of $Nd^{3+}$ (at 866 nm) in $Nd^{3+}$ (0.5%), $Yb^{3+}$ (1, 2, 3, and 5%):$LiYF_4$ phosphors. The excitation wavelength $\lambda_{ex}$ = 355 nm corresponds to $^4I_{9/2}$–$^4D_{5/2}$ absorption band of $Nd^{3+}$. The inset demonstrates the luminescence rising time as a function of the $Yb^{3+}$ concentration. (**b**). Normalized room temperature lifetime curves of $^4F_{3/2}$–$^4I_{9/2}$ emission of $Nd^{3+}$ (at 866 nm) in $Nd^{3+}$ (0.5%), $Yb^{3+}$ (0, 1, 2, 3, and 5%):$LiYF_4$ phosphors. The excitation wavelength $\lambda_{ex}$ = 520 nm corresponds to $^4I_{9/2}$–$^2K_{13/2}$/$^2G_{9/2}$ absorption band of $Nd^{3+}$.

It was important to ensure that the samples were not heated up by the excitation irradiation. We selected the well suited excitation power densities according to [15]. The temperature evolution of the normalized luminescence spectra of $Nd^{3+}$ (0.5%), $Yb^{3+}$ (2.0%):$LiYF_4$ in the 80–320 K range is shown in Figure 6. It can be seen that the relative intensities of the $Nd^{3+}$ and $Yb^{3+}$ emission peaks depended on the temperature for all samples. In order to investigate this spectral temperature-dependence, the luminescence intensity ratio (LIR) was calculated according to the common Equation (1). The LIR values were carried out in the wavelength range of 842–922 nm ($Nd^{3+}$) and 936–1037 nm ($Yb^{3+}$) for both 355 and 520 nm excitations.

$$\text{LIR(Y)} = \frac{\int I_{Nd}(\lambda, T)d\lambda}{\int I_{Yb}(\lambda, T)d\lambda} \tag{1}$$

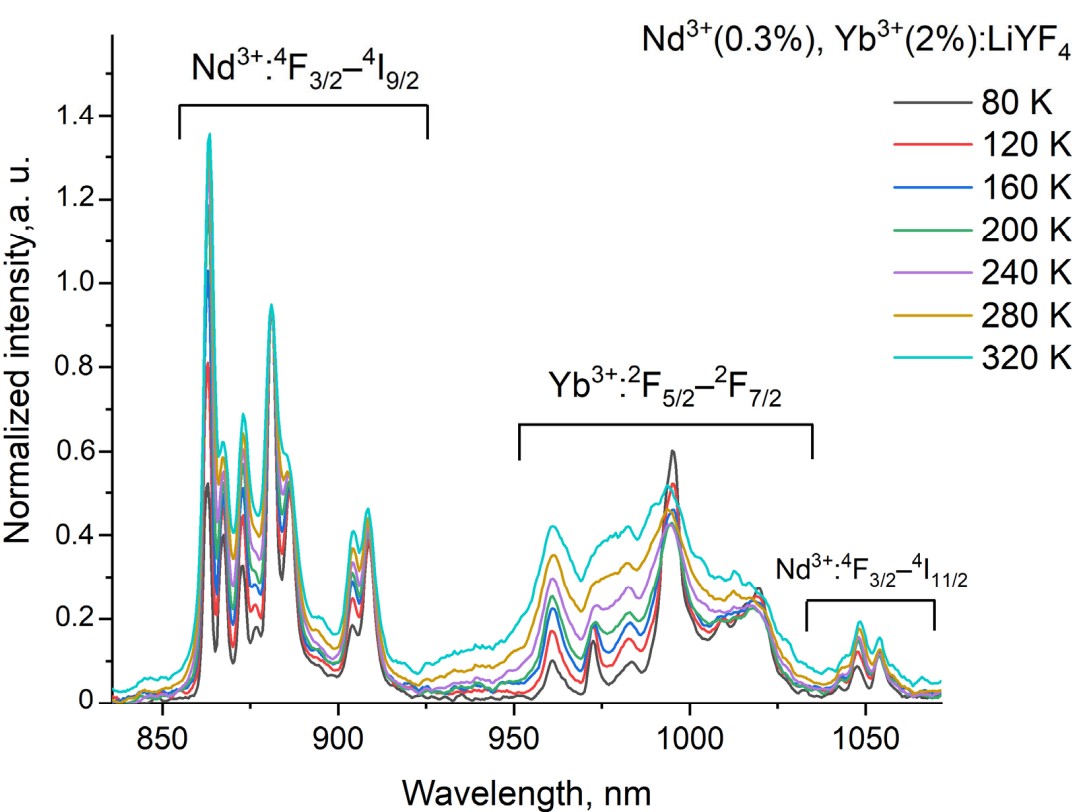

**Figure 6.** The normalized at 881 nm $Nd^{3+}$ (0.3%), $Yb^{3+}$ (2.0%):$LiYF_4$ luminescence spectra recorded in the 80–320 K range. The excitation $\lambda_{ex}$ = 520 nm corresponds to $^4I_{9/2}$–$^2K_{13/2}$/$^2G_{9/2}$ absorption band of $Nd^{3+}$.

The LIR dependences are presented in Figure 7. It is clearly seen that the shape of the LIR functions demonstrated weak dependence on the concentration of $Yb^{3+}$ at a constant 0.3 mol.% of $Nd^{3+}$. The LIR function had complex temperature dependence with segments of an increase in the range of 77–220 K, as well as a decrease character in the range of 230–320 K. This dependence was most pronounced for 1 mol.% $Yb^{3+}$. Based on the fact that the nature of the change in the LIR function also depended on the processes of energy transfer between two ions, there is a possibility that cross-relaxation influenced the character of the change in the LIR curves. The LIR functions of the $Nd^{3+}$ (0.3%), $Yb^{3+}$ (1.0–5.0 mol.%):$LiYF_4$ samples had complicated shape with growth part in the temperature range of 75–200 K and a decay part in the temperature range of 200–320 K. The growing character of the LIR function takes place because $Nd^{3+}$ intensity increases faster that $Yb^{3+}$ one with temperature. It occurred because the phonon appearance probability is relatively low in this temperature range. Then, this probability increased and the population of

$^2F_{5/2}$ level of $Yb^{3+}$ happened more efficiently and the intensity of $Yb^{3+}$ started to grow, in turn the efficiency of depopulation of $^4F_{3/2}$ level of $Nd^{3+}$ also increased and $I_{Nd}/I_{Yb}$ started to decay with the temperature increase. Figure 8 shows the LIR curves at different excitation wavelengths of 355 and 520 nm for the sample $Nd^{3+}$ (0.3%), $Yb^{3+}$ (2%): $LiYF_4$. As we can see in the Figure 8, the shape of the LIR curves as a function of temperature is independent of excitation wavelengths. To quantitatively compare both functions, we represented the parameters of fitting procedure in Table S1 of Supplementary File. The parameters are very close to each other. This may tell us that the population of the $Yb^{3+}$ level due to the cross-relaxation process does not notably affect the temperature sensitivity and the efficiency of cross-relaxation does not depend on temperature. However, it is worth considering that the same LIR value was achieved for two different temperatures for 120 and 280 K. It is worth considering that temperature measurement for this class of substances was possible from 200 K., which is suitable for measurements in fundamental biology, circuitry, etc. [25].

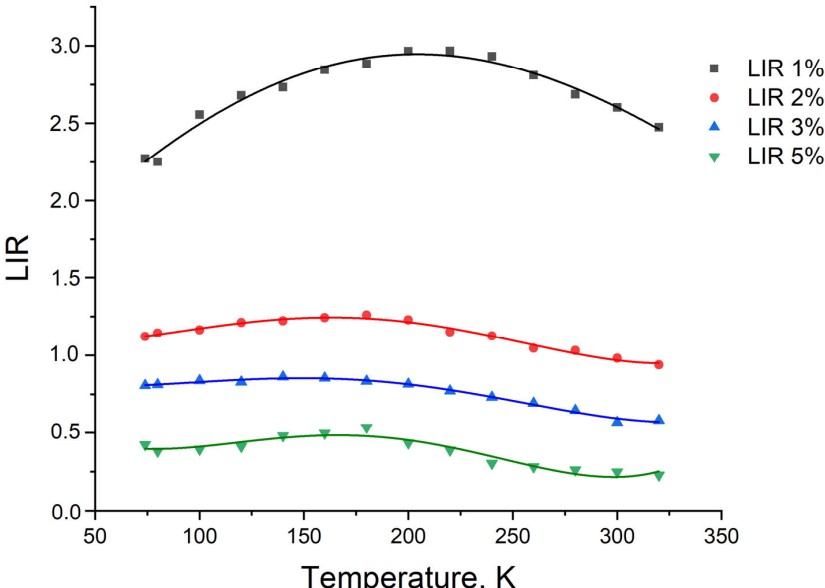

**Figure 7.** The LIR curves for I ($Nd^{3+}$, $^4F_{3/2}$–$^4I_{9/2}$, ~866 nm) and I ($Yb^{3+}$, $^2F_{5/2}$–$^2F_{7/2}$, ~980 nm) luminescence peaks.

In order to strengthen the argument of the low temperature dependence of cross-relaxation in the studied system, we performed kinetic characterization of the studied samples in the 80–320 K range. Indeed, the presence of rising time curve for $^4F_{3/2}$–$^4I_{9/2}$ peak at 355 nm excitation was related to non-resonant excitation. The $^4F_{3/2}$ is populated via transitions from the higher levels that have their own lifetime of the exited state. Thus, rising time was determined by the rate of transitions from higher levels as well as the rate of cross-relaxation [19,26]. If the rising time depends on temperature, hence, one or both above-mentioned processes are temperature dependent. The results of kinetic characterization of $Nd^{3+}$ (0.3%), $Yb^{3+}$ (2.0%):$LiYF_4$ sample under 355 nn excitation are represented in Figure 9.

It can be seen that rising time did not demonstrate clear temperature dependence. The obtained results are an argument for the low temperature dependence of cross-relaxation process at least in the precent temperature range. It can also be observed that in the 80–200 K range, the decay curves were almost the same and in the higher temperature range the decay time decreased with the increase in temperature. This phenomenon cab be explained by the fact that the energy transfer between $^4F_{3/2}$ ($Nd^{3+}$) and $^2F_{5/2}$ ($Yb^{3+}$) was phonon-assisted and the probability of phonon appearance increased with the temperature increase.

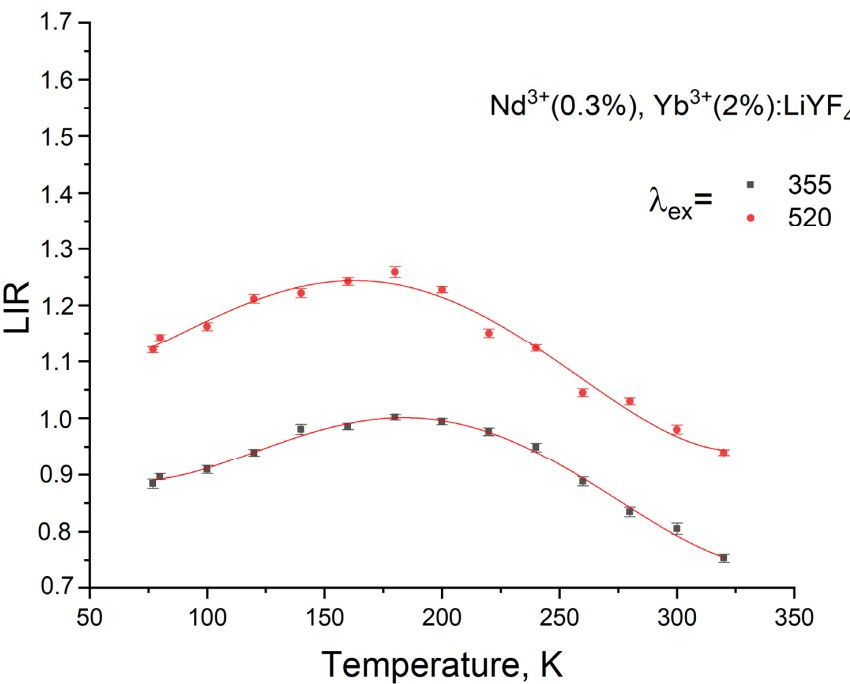

**Figure 8.** The LIR curves of both I(Nd$^{3+}$, $^4F_{3/2}$–$^4I_{9/2}$, ~866 nm) and I (Yb$^{3+}$, $^2F_{5/2}$–$^2F_{7/2}$, ~980 nm) luminescence peaks at different excitation wavelengths λex = 520 nm and λex = 355 nm.

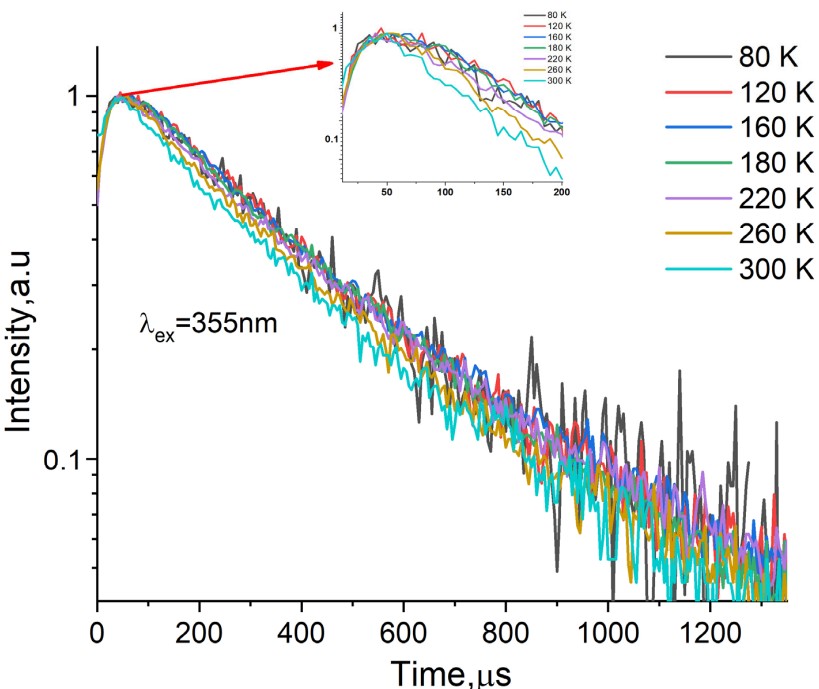

**Figure 9.** Lifetime curves of Nd$^{3+}$ (0.3%), Yb$^{3+}$ (2.0%):LiYF$_4$ sample under 355 nm excitation in the 80–320 K range.

For temperature mapping instrumentation, the absolute S$_a$ and relative temperature sensitivity S$_r$ are very crucial performances. The S$_a$ [K$^{-1}$] and S$_r$ [%*K$^{-1}$] are determined as [2,27]:

$$S_a = \frac{d(\text{LIR})}{dT} \qquad (2)$$

$$S_r = \frac{1}{LIR}\left|\frac{d(LIR)}{dT}\right| * 100\% \tag{3}$$

The $S_r$ and $S_a$ curves for all the samples are represented in Figure 10a,b, respectively. The maximum $S_a$ value is achieved for $Nd^{3+}$ (0.3%), $Yb^{3+}$ (1.0%):$LiYF_4$ ($S_a$ = 0.007 $K^{-1}$ at 320 K). The rest of the samples demonstrate notably lower values of $S_a$. The maximum $S_r$ value is achieved for $Nd^{3+}$ (0.3%), $Yb^{3+}$ (5.0%):$LiYF_4$ ($S_r$ = 1.03%*$K^{-1}$ at 260 K). The $S_a$ and $S_r$ values of the studied samples are compared to the counterparts in Table 1. Our samples demonstrate competing values compared to the temperature sensitivity values for these substances.

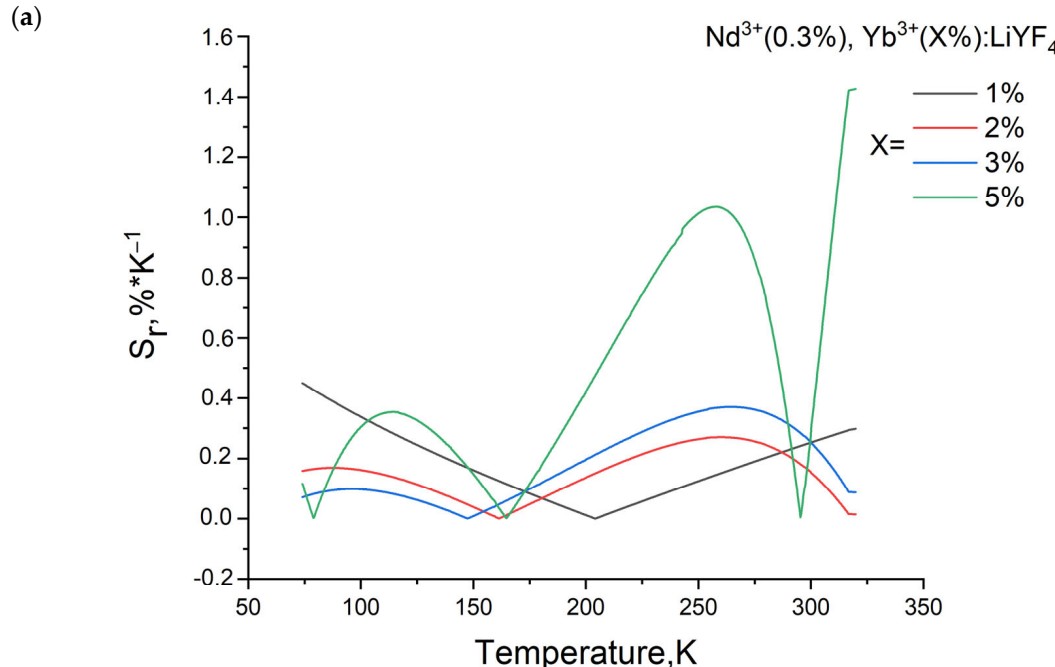

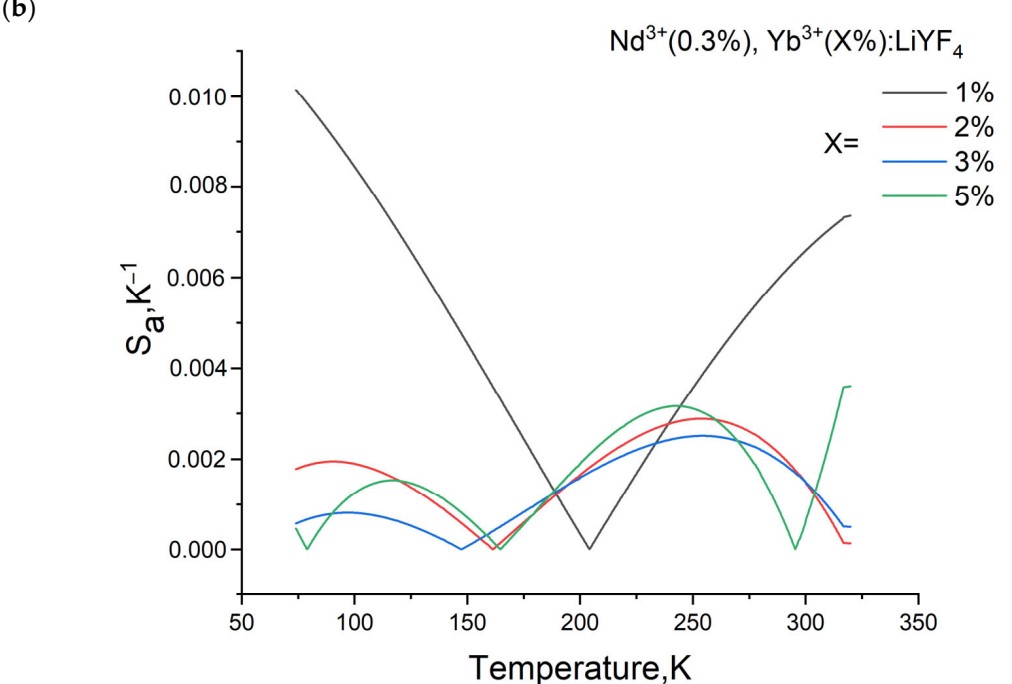

**Figure 10.** (**a**) Relative temperature sensitivity ($S_r$) and (**b**) absolute temperature sensitivity ($S_a$) as a function of temperature for $Nd^{3+}$, $Yb^{3+}$:$LiYF_4$ samples.

**Table 1.** The performances of rare-earth doped optical thermometers.

| Sample | Transitions, Detected Wavelengths, and Conditions of the Excitation | Maximum $S_a$ [$K^{-1}$] | Maximum $S_r$ [%/K] | T,K | Ref. |
|---|---|---|---|---|---|
| $Tb_{0.99}Eu_{0.01}(BDC)_{1.5}(H_2O)_2$ | $Eu^{3+}$ ($^5D_0$–$^7F_2$), $Tb^{3+}$ ($^5D_4$–$^7F_5$) $\lambda_{ex}$ = 320 nm | - | 0.14 | 283–333 | [28] |
| $LiYF_4$: $Nd^{3+}$, $Yb^{3+}$ | $Nd^{3+}$ ($^4F_{3/2}$–$^4I_{9/2}$), $Yb^{3+}$ ($^4F_{5/2}$–$^2F_{7/2}$), $\lambda_{ex}$ = 520 nm | 0.007 | 1.03 | 240–320 | This work |
| $SrTiO_3$:$Ni^{2+}$,$Er^{3+}$ | $Er^{3+}$ ($^4I_{13/2} \rightarrow {}^4I_{15/2}$) $Ni^{2+}$ ($^3T_{2g}(F) \rightarrow {}^3A_{2g}(F)$), $\lambda_{ex}$ = 375 nm | - | 0.76 | 303 | [29] |
| $YVO_4$:$Nd^{3+}$ | $Nd^{3+}$ ($^4F_{3/2}$–$^4I_{11/2}$) $\lambda_{ex}$ = 808 nm | - | 0.46 | 323 | [16] |
| $PrP_5O_{14}$ | $Pr^{3+}$ ($^3P_0 \rightarrow {}^1D_2$) $\lambda_{ex}$ = 488 nm | - | 0.46 | 363 | [30] |
| $NaPr(PO_3)_4$ | $Pr^{3+}$ ($^3P_0$ –$^3H_6$) $\lambda_{ex}$ = 488 nm | 0.0043 | - | 300–365 | [30] |
| $LaF_3$:$Nd^{3+}$ | $Nd^{3+}$ ($^4F_{3/2}$ -$^4I_{9/2}$) $\lambda_{ex}$ = 808 nm | - | 0.1 | 293 | [31] |
| MOF: $Eu^{3+}$/$Tb^{3+}$ | $Tb^{3+}$ ($^5D_4 \rightarrow {}^7F_5$) and $Eu^{3+}$ ($^5D_0 \rightarrow {}^7F_2$) $\lambda_{ex}$ = 340 nm | - | 0.57 | 150–300 | [32] |
| $YF_3$: $Nd^{3+}$, $Yb^{3+}$ | $Nd^{3+}$ ($^4F_{5/2}$–$^4I_{11/2}$) $\lambda_{ex}$ = 790 nm | 0.64 | 0.92 | 100 | [33] |
| $NaYbF_4$: Nd@$NaYF_4$: Nd | $Nd^{3+}$ ($^4F_{5/2}$–$^4I_{11/2}$) $\lambda_{ex}$ = 790 nm | 0.7 | | 300 | [34] |

## 4. Conclusions

Potential optical temperature sensors based on $Nd^{3+}$ (0.3 mol.%), $Yb^{3+}$ (1.0, 2.0, 4.0, and 5.0 mol.%):$LiYF_4$ phosphors were studied in the 80–320 K range. It was found that the LIR function depends on the $Yb^{3+}$ concentration at a fixed 0.3 mol.% of $Nd^{3+}$. The LIR functions were studied at different excitation wavelengths at 355 and 520 nm. It was shown that the cross-relaxation does not have a notable impact on the temperature sensitivity of the spectral characteristics of the samples. The maximum value of $S_a$ is achieved for $Nd^{3+}$ (0.3%), $Yb^{3+}$ (1.0%): $LiYF_4$ ($S_a$ = 0.007 $K^{-1}$ at 320 K) and $S_r$ $Nd^{3+}$ (0.3%), $Yb^{3+}$ (5.0%): $LiYF_4$ ($S_r$ = 1.03%*$K^{-1}$ at 260 K).

**Supplementary Materials:** The following supporting information can be downloaded at: https://www.mdpi.com/article/10.3390/photonics10040375/s1. Figure S1. Experimental set-up. 1 and 4 are lenses, 2 and 3 are crio-system and the sample, 5–optical filter, 6–waveguide, 7–spectrometer, 8–PC. Table S1. Polynomal parameters for LIR

**Author Contributions:** A.G.: investigation, data curation, writing—original draft, M.P.: conceptualization, investigation, resources, writing—original draft, E.O.: investigation, S.K.: investigation, O.M.: investigation. All authors have read and agreed to the published version of the manuscript.

**Funding:** The study was funded by the grant from the Russian Science Foundation number 22-72-00129, https://rscf.ru/project/22-72-00129/ (accessed on 1 August 2022).

**Institutional Review Board Statement:** Not applicable.

**Informed Consent Statement:** Not applicable.

**Data Availability Statement:** Not applicable.

**Conflicts of Interest:** The authors declare no conflict of interest.

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
