# Peer review of "Optical Temperature Sensors Based on Down-Conversion Nd3+,Yb3+:LiYF4 Microparticles"

_photonics, doi:10.3390/photonics10040375_

Round 1

Reviewer 1 Report

Title: Optical temperature sensors based on down-conversion Nd3+,Yb3+:LiYF4 microparticles 

Authors: Anna Ginkel  et al.

The work investigates the impact of excitation conditions on temperature sensitivity of Nd3+,Yb3+:LiYF4 phosphors. 

The luminescence intensity ratio (LIR) of Nd3+ ~866 nm and Yb3+ ~980 nm emissions chosen as a temperature-dependent parameter. Green (520 nm) and ultraviolet (355 nm) excitation channels are used. Depending on the excitation wavelength, the energy has differents pathways to be transfered from Nd3+ and Yb3+. One of them occurs via phonons (which explains the temperature dependence). 

Author fixed Nd3+ concetration and study the dependence of LIR function on the Yb3+ concentration. They obtain the absolute and relative sensitivity parameters and compares their values with other reported data.

The summary is reasonable (improvable) and the structure of the document is adequate, although the justification of cross-relaxation as a process (lines 105-127) could be improved and some of the reasoning could be refuted or due to other processes.

I understand that

line 148-149 "The LIR function has complex temperature dependence with segments of an increase in the range of 77–220 K as well as a decrease character in the range of 230–320 K."

In following lines authors suspects that ..."the nature of the change in the LIR function" was dependent ... "on the processes of energy transfer between the two ions" involved in the procese.

Thus authors argued ...  "that temperature measurement for this class of substances is possible from 200 K" up to 320 K, and discard the its use of this sistem at temperatures below.

After that they obtain the absolute and also the relative temperature sensitivity (Sa [K− 1] and Sr [%*K− 1]) and report their reults in fig 7.

In this figure that includes the entire range of T (Why? The authors have just rejected the use of this system in the range of 77 to 200K). The figure is confusing by having multiple lines whose behavior is too different.

A certain trend is apparent if we disregard the data for the 1% sample and focus on the 160-320K range. But the behavior of the 5% sample is also so different that is not clear what is the relevant information in this figure.

In addition the paragraph in lines 170-180 does not clarify what the authors mean.

The only two relevand data are 

The maximum Sa value is achieved for Nd3+ (0.3%), Yb3+ (1.0%):LiYF4 (Sa = 0.007 K−1 at 320 K)

The maximum Sr value is achieved for Nd3+ (0.3%), Yb3+ (5.0%):LiYF4 (Sr = 1.03%*K−1 at 260 K).

What about reproducibility? Why is the 5% curve in fig 7 (particularly 7a) so different? This facts are important because 1/ this curve (5%) has no sensitivity at 300 K and 2/ from figure 7, the repeatability seems assured in samples in which the concentration is in the range of 2-3% because of curves have similar behaviour, but can readers' understand that repeatability is not assured with samples in which the concentration is outside of this range?

And finally, if the maximum values for Sa and Sb are obtained in the samples that do not show a sistematic trend, what is the true meaning of this values?

Unfortunately, in my oppinion, the text lacks adequate scientific quality. I understand that some results are difficult to clearly stated, but in this work most of the explanations are doubtfull, or at least not clear enough.

Due to this, I believe that manuscript requires important changes and clarifications before publication and needs a thorough review by the authors.

Two additional remarks

1/ Note the change of font size in lines 159-168

2/ Refs. [8],[15] and [27] refers to the same article

   Bednarkiewicz et al, Physical Chemistry Chemical Physics, 2015; Volume 17, pp. 24315-24321.

    Refs. [17],[19] and [21] also refers to the same article

   Fedorov et al, Inorganic Materials, 2022; Volume 58, pp. 223-245

Check carefully the references to avoid this kind of errors.

Reviewer 2 Report

A manuscript by Anna Ginkel et al reports on the study of the luminescence properties of LiYF4 crystals co doped with Nd and Yb ions. Authors present the data on the modification of luminescence spectra as a function of Yb concentration as well as decay curves of emission. Studies were performed under excitation with two excitation wavelength, which allows to study the contribution of cross relaxation process on the dependence of Yb concentration. Temperature sensitivity of the sudied crystals was estimated as well. I may propose this manuscript for publication after revision according to the following remarks.

1.      Abstract contains puzzling data and should be carefully corrected. Currently it reports that the studies were performed also for the samples with 0.5 mol% Nd that contradicts to the data presented in the manuscript. Different data on the temperature sensitivity are presented for the samples with Nd concentration 0.3 and 0.5 mol% while the data for the latter samples were not presented in the manuscript.

2.      Page 2. Line 87. You mention that under 355 nm excitation relaxation occurs via cross relaxation and radiative transitions from the higher energy levels. Which concrete radiative transitions do you mean? Please present the luminescence spectra demonstrating these radiative transitions.

3.      Fig.1 contains XRD pattern as well as energy level scheme that is confusing why so different data are presented as one figure. I propose to split it to two separate figures. Moreover, the scheme is presented with low resolution and its quality should be improved.

4.      Page 4, line 126. Please correct misprint. “It can also be seen that, that cross-relaxation…”

5.      Figures 5 and 6 contain data on the integrated luminescence intensity. In which region did you integrate the signal and why did you choose this region?

6.      Page 5, lines 158-161. Discussion does not correspond to the presented data. In previous sentences you discuss the data obtained for the sample with 2% of Yb. LIR values at 120 and 280 K are different for this sample while it is written here that the values are the same. Please change this part.

7.      References should be revised and formatted in unique style according to the journal rules.

Reviewer 3 Report

The manuscript presents an application of Nd3+, Yb3+ 3:LiYF4 nanoparticles as temperature sensors. The authors evaluated the impact of the metal ion's composition and cross-relaxation on the photoluminescence properties of these nanoparticles. However, there are several areas that can be further improved to enhance the overall quality of the manuscript, and I recommend major revisions addressing the following comments:

The significance of this research needs to be stated more clearly. To improve the clarity of the research's significance, the authors can delve deeper into one or both of the following aspects:

1.     The impact of cross-relaxation on this type of phosphor needs a better elucidation. On line 40, it is mentioned that the temperature-dependence is questionable. Please cite previous studies that support this claim. Additionally, it would be helpful to provide the rate equation of cross-relaxation to give readers a clear physical insight. Instead of merely stating that temperature sensitivity is not affected by cross-relaxation, the authors should propose a mechanism explaining why it does not change the T-dependence. For example, it could be because other energy transfers dominate the photoluminescence process over cross-relaxation due to higher rates, or cross-relaxation could be temperature-dependent as well. Lastly, the authors should explain why cross-relaxation is relevant since they can use redder excitation to avoid cross-relaxation.

2.     A more rigorous demonstration of Nd3+, Yb3+ 3:LiYF4 nanoparticles' application on temperature sensor is necessary. The authors should complete the integral and differential operations in equation (1), (2), and (3). Once a specific relation between LIR and T is obtained, the authors should fit the experimental data with the derived equation. (See previous studies for reference: Wei, Hongling, et al. "Nd3+-sensitized NIR downshifting emission in NaYbF4: Nd@ NaYF4: Nd nanoparticles for deep tissue temperature sensing." Optical Materials 124 (2022): 112016.) A high R-value from the fitting would be strong evidence of a legitimate temperature sensor. The motivation for using Nd3+, Yb3+ 3:LiYF4 microparticles can also be improved. The authors should explain why these materials are special compared to other compounds with rare earth metals. They should also justify why research on down-conversion is necessary, rather than up-conversion, which has already been studied and reported. See:

Zhao, Yan, et al. "Optical temperature sensing of up-conversion luminescent materials: Fundamentals and progress." Journal of Alloys and Compounds 817 (2020): 152691.

Before concluding that cross-relaxation does not change the game, more evidence is needed. Two LIR-T curves with similar shapes are not sufficient. The authors should fit the two LIR curves with different excitation wavelengths, compare the fitting parameters, and calculate Sa and Sr to evaluate the impact of cross-relaxation.

The authors should also address the following minor issues:

1.     Please provide details about the sample synthesis, such as where the chemicals were purchased.

2.     Please provide the model of your JV LOTIS laser, e.g., LS-2138-30T.

3.     Please consider plotting the experimental setup.

4.     The resolution of figure 1.b. is low.

5.     Please use the full name of LIR on line 138 where it is first mentioned.

6.     Please repeat the temperature-dependence experiments for LIR and add error bars to all the LIR-T plots.

Round 2

Reviewer 1 Report

As with all manuscripts, it is subject to changes and could be improved, but I believe that the changes already made significantly improve the work and make it easier to read and understand.

I recommend the publication of this work in its present form. 

Reviewer 2 Report

The authors carefully modified the manuscript according to the presented remarks. I can recommend this manuscript for publication.

Reviewer 3 Report

The revisions are sufficient. I have no further comments